# In Utero Exposure to Caffeine and Acetaminophen, the Gut Microbiome, and Neurodevelopmental Outcomes: A Prospective Birth Cohort Study

**DOI:** 10.3390/ijerph19159357

**Published:** 2022-07-30

**Authors:** Hannah E. Laue, Yike Shen, Tessa R. Bloomquist, Haotian Wu, Kasey J. M. Brennan, Raphael Cassoulet, Erin Wilkie, Virginie Gillet, Anne-Sandrine Desautels, Nadia Abdelouahab, Jean Philippe Bellenger, Heather H. Burris, Brent A. Coull, Marc G. Weisskopf, Wei Zhang, Larissa Takser, Andrea A. Baccarelli

**Affiliations:** 1Department of Epidemiology, Geisel School of Medicine, Dartmouth College, Hanover, NH 03755, USA; 2Department of Environmental Health Sciences, Columbia University Mailman School of Public Health, New York, NY 10032, USA; ys3419@cumc.columbia.edu (Y.S.); tb2715@cumc.columbia.edu (T.R.B.); hw2694@cumc.columbia.edu (H.W.); kaseyjmbrennan@gmail.com (K.J.M.B.); andrea.baccarelli@columbia.edu (A.A.B.); 3Département de Chimie, Université de Sherbrooke, Sherbrooke, QC J1K 2R1, Canada; raphael.cassoulet@hotmail.fr (R.C.); jean-philippe.bellenger@usherbrooke.ca (J.P.B.); 4Département de Pédiatrie, Université de Sherbrooke, Sherbrooke, QC J1K 2R1, Canada; erin.wilkie@usherbrooke.ca (E.W.); virginie.gillet@usherbrooke.ca (V.G.); infoasdpsy@gmail.com (A.-S.D.); nadia.abdelouahab@usherbrooke.ca (N.A.); larissa.takser@usherbrooke.ca (L.T.); 5Department of Pediatrics, Perelman School of Medicine, University of Pennsylvania, Philadelphia, PA 19104, USA; burrish@chop.edu; 6Division of Neonatology, Department of Pediatrics, Children’s Hospital of Philadelphia, Philadelphia, PA 19104, USA; 7Department of Biostatistics, Harvard T.H. Chan School of Public Health, Boston, MA 02115, USA; bcoull@hsph.harvard.edu; 8Department of Environmental Health, Harvard T.H. Chan School of Public Health, Boston, MA 02115, USA; mweissko@hsph.harvard.edu; 9Department of Preventive Medicine, Northwestern University Feinberg School of Medicine, Chicago, IL 60611, USA; wei.zhang1@northwestern.edu; 10Département de Psychiatrie, Faculté de Médicine et Sciences de la Santé, Université de Sherbrooke, Sherbrooke, QC J1K 2R1, Canada

**Keywords:** acetaminophen, caffeine, microbiome, neurodevelopment, children’s health

## Abstract

Pregnant individuals are exposed to acetaminophen and caffeine, but it is unknown how these exposures interact with the developing gut microbiome. We aimed to determine whether acetaminophen and/or caffeine relate to the childhood gut microbiome and whether features of the gut microbiome alter the relationship between acetaminophen/caffeine and neurodevelopment. Forty-nine and 85 participants provided meconium and stool samples at 6–7, respectively, for exposure and microbiome assessment. Fecal acetaminophen and caffeine concentrations were quantified, and fecal DNA underwent metagenomic sequencing. Caregivers and study staff assessed the participants’ motor and cognitive development using standardized scales. Prenatal exposures had stronger associations with the childhood microbiome than concurrent exposures. Prenatal acetaminophen exposure was associated with a trend of lower gut bacterial diversity in childhood [β = −0.17 Shannon Index, 95% CI: (−0.31, −0.04)] and was marginally associated with differences in the relative abundances of features of the gut microbiome at the phylum (Firmicutes, Actinobacteria) and gene pathway levels. Among the participants with a higher relative abundance of Proteobacteria, prenatal exposure to acetaminophen and caffeine was associated with lower scores on WISC-IV subscales. Acetaminophen during bacterial colonization of the naïve gut is associated with lasting alterations in childhood microbiome composition. Future studies may inform our understanding of downstream health effects.

## 1. Introduction

Acetaminophen, the only analgesic and antipyretic recommended during pregnancy, is used by at least two-thirds of pregnant people and is also one of the most frequently used medications in childhood [1,2,3]. Recent evidence suggests potential adverse neurodevelopment resulting from exposure, including impaired cognition and increased risk of attention deficit-hyperactivity disorder [4,5], although findings are inconsistent [6,7]. Similarly, physicians recommend that caffeine intake in pregnancy be limited to less than 200 mg per day [8]. This recommendation is partially due to evidence from animal models suggesting that high-dose prenatal exposure causes decreased brain weight, neural tube and central nervous system defects, and adverse behavioral outcomes [9,10,11], but epidemiologic human studies are inconclusive [12,13,14]. Studies of childhood caffeine consumption are limited, but it is estimated to average 0.3–1 mg/kg/day in the United States [15].

Factors that confer resilience or susceptibility to these exposures may explain discrepancies in prior findings but have been understudied. One such factor is the gut microbiome, the myriad microorganisms that reside in the gastrointestinal tract [16]. Some bacteria contribute to the metabolism of potentially neurotoxic compounds, including acetaminophen [17], while others may alleviate harmful effects by producing beneficial compounds. In contrast, pathogens may exacerbate adverse associations. By examining microbial features that alter the relationships of caffeine and acetaminophen with neurodevelopment, we may begin to understand the inconsistencies of prior studies. Further, we may identify bacterial species or byproducts that can be used therapeutically to reduce adverse effects when exposure is unavoidable due to limited antipyretic options in pregnancy.

Bacteria colonize the naïve infant gut after birth, and their ability to establish communities is affected by the environment they experience, including chemical exposures [18,19,20]. Acetaminophen and caffeine cross the placenta [21,22,23], undergo limited metabolism by the fetus [24,25], and thus accumulate in meconium, making it an ideal matrix to assess exposure experienced by early colonizers of the gut microbiome. Although the infant gut microbiome is highly variable, early exposures can have lasting impacts that are detectable into childhood [26]. Additionally, with the ingestion of acetaminophen and caffeine continuing into childhood, stool remains an informative exposure matrix for microbiome studies. Few studies have examined whether either acetaminophen or caffeine at any age is associated with changes in the microbiome, and none have examined prenatal exposure [27,28,29,30].

We aimed to understand the role of the childhood gut microbiome in the associations of two of the most common medications/exposures in pregnancy, acetaminophen, and caffeine, with neurodevelopment in the GESTation and Environment (GESTE) cohort.

## 2. Materials and Methods

### 2.1. Study Participants

Patients were recruited to the GESTE cohort in early pregnancy or at the time of birth between 2007 and 2009 at the University of Sherbrooke, Quebec, Canada (cohort description Appendix A). When the children were aged 6–7 years old, they underwent a battery of neurocognitive tests. In addition, a convenient sample of children provided stool samples for microbiome analysis. This analysis included children with stool samples at age 6–7 that underwent metagenomic sequencing, completed the neurocognitive tests, and had exposures assessed in the meconium (prenatal, n = 49) and/or stool at age 6–7 (cross-sectional, n = 85). Written informed consent was obtained from parents. All of the study protocols were approved by the Institutional Review Boards of the University of Sherbrooke, Harvard T.H. Chan School of Public Health, and Columbia University.

### 2.2. Exposure Assessment

Meconium was collected from infants who passed it after birth at the hospital and stored at −80 °C, and the stools were collected by caregivers at the participants’ homes when they were 6–7 years old. Caffeine and acetaminophen were extracted from the feces and quantified as previously described [6,31]. Caffeine exposures were log_2_ transformed for all analyses to reduce the influence of outlier concentrations. Approximately half of the meconium and 90% of the childhood stool had non-detectable exposure to acetaminophen; thus, we dichotomized exposure as exposed (detected) and non-exposed (below the limit of quantification, LOQ). The details on the LOQ for both exposures are available in the Appendix A.

### 2.3. Microbiome Assessment

Total DNA was extracted from childhood stool using established protocols [32]. Metagenomic sequencing was conducted at New York University’s Langone Genome Technology Center. After preprocessing and quality control as previously described [32], bacterial species and their pathways were annotated with MetaPhlAn 3 and HUMAnN 3 (MetaCyc) [33], respectively, and normalized to relative abundance tables (i.e., divided by the total number of reads for each sample) [34]. Bacterial alpha diversity was quantified with the Shannon [35] and Pielou [36] Indices, calculated from species relative abundance tables with the *phyloseq* package [37]. Species beta diversity was measured with UniFrac (weighted and unweighted) [38], Bray-Curtis [39], and Jaccard [40] distances.

### 2.4. Neurodevelopmental Assessment

When the participants were 6–7 years old, caregivers completed the Questionnaire sur le Trouble de L’Acquisition de la Coordination (QTAC), the French equivalent of the Developmental Coordination Disorder Questionnaire. Through this, they compared their child’s motor development to their peers [41]. In addition, a trained member of the study staff administered five subtests from the Wechsler Intelligence Scale for Children, 4th edition (WISC-IV) to the children, including Block Design, Coding, Digit Span, Information, and Vocabulary [42]. To assess an underlying component of intelligence not captured by individual subtests, we created a summary score of the five completed subtests (sum of five subtests; WISCsum).

### 2.5. Statistical Methods

The covariates were selected based on their potential to confound the association between the exposures and the microbiome based on prior knowledge. In final analyses, we adjusted for sex, ever breastfeeding, mode of birth (vaginal or caesarean), and family income (Appendix A). In sensitivity analyses, we adjusted for maternal IQ (Appendix A) and for both exposure windows concurrently. The alpha diversity indices were linearly regressed against caffeine or acetaminophen, adjusting for covariates. A Benjamini–Hochberg false discovery rate (FDR) of *q* < 0.1 was considered statistically significant [43]. Differences in beta diversity related to caffeine and acetaminophen were tested with the *adonis* function, a permutational analysis of variance [44]. Microbiome Multivariable Association with Linear Models (MaAsLin2) was used to determine whether acetaminophen or caffeine showed a trend of association (*q* < 0.1) with bacterial phyla, species, or pathways with a prevalence above 10% [45].

We have previously reported no association between acetaminophen exposure and QTAC/WISC-IV scores [6], but we hypothesized that features of the microbiome might modify the relationships between prenatal acetaminophen or caffeine and neurodevelopment. To limit multiple testing penalties, we screened the most abundant (top 10%) bacterial species (n = 34), pathways (n = 36), and all phyla (n = 9) for interactions with caffeine and acetaminophen using linear regression. Model fit was assessed using likelihood ratio tests. All of the statistical analyses were conducted in the R statistical environment (v4.0.2; R Core Team, Vienna, Austria). Code is available at https://github.com/YikeShen (accessed on 27 June 2022).

## 3. Results

The population characteristics among the participants included in the analysis were similar to the full cohort examined at age 6–7 years (Table 1). The GESTE cohort is relatively homogeneous in demographics (primarily upper-middle-class French-Canadians). The mean maternal age at recruitment was 29 years, and more than half of the participants were nulliparous. Breastfed children (87.8%) were overrepresented in the analytical population. The average family income was higher at follow-up than at delivery, but our analytical cohorts were similar to the full cohort. The percentage of the participants with detectable levels of acetaminophen in their meconium was similar in the overall and analytical populations (Table 1). All of the participants had detectable levels of caffeine in the meconium, with a median concentration of 390 ng/g meconium in the pilot population compared to 399 ng/g meconium in all children with available meconium who were followed to an age of 6 years (Table 1). The median caffeine concentration was significantly lower in childhood (24 ng/g stool, 100% detection), and fewer (10%) participants had detectable levels of acetaminophen (Table 1).

Prenatal acetaminophen exposure was associated with lower alpha diversity using multiple indices [e.g., β = 0.17 lower Shannon Index comparing exposed to unexposed (95% CI: −0.21, −0.09), *q* = 0.044; Figure 1, Appendix A. At the phylum level, prenatal acetaminophen exposure showed a trend of association with lower relative abundance of Firmicutes [β = −0.09 comparing exposed to unexposed (95% CI: −0.16, −0.03), *q* = 0.062] and higher relative abundance of Actinobacteria [Figure 2, Appendix A, β = 0.183, (95% CI: 0.03, 0.34), *q* = 0.085], but there were no associations at the species level (Appendix A). In sensitivity analyses co-adjusting for cross-sectional exposure, the associations were of a similar magnitude (Appendix A). The microbiomes of children exposed to prenatal acetaminophen had higher relative abundance of genes in the succinate fermentation to butanoate pathway [Figure 3, Appendix A, MetaCyc Pathway ID: PWY-5677; β = 0.33, (95% CI: 0.17, 0.49), *q* = 0.072]. In contrast, prenatal caffeine exposure was not significantly associated with alpha diversity or the relative abundance of bacterial species, phyla, or functional pathways (Figure 1 and Figure 3, Appendix A). Neither prenatal exposure was associated with differences in beta diversity (Appendix A). In general, cross-sectional exposure was not associated with differences in the microbiome, even when restricting to subjects with meconium exposures, although caffeine exposure in stool at age 6–7 years was associated with an increased relative abundance of three pathways (Appendix A): methionine synthesis [MetaCyc Pathway ID: PWY-5345; β = 0.24 (0.12, 0.35), *q* = 0.018], assimilatory sulfate reduction [MetaCyc Pathway ID: SO4ASSIM-PWY; β = 0.24 (0.12, 0.36), *q* = 0.018], and sulfate assimilation and cysteine synthesis [MetaCyc Pathway ID: SULFATE-CYS-PWY; β = 0.24 (0.12, 0.35), *q* = 0.018].

When testing whether the most abundant phyla, species, or pathways modified the association between exposure to acetaminophen/caffeine and neurodevelopmental scores, we found that the relative abundance of Proteobacteria modified the neurodevelopmental effects of both acetaminophen and caffeine (Figure 4). The participants with detectable concentrations of acetaminophen had slightly higher scores on WISC-IV subtests, as previously reported, [6] but this benefit was weakened among those with higher levels of Proteobacteria (Appendix A). In contrast, higher concentrations of caffeine related to lower scores on several WISC-IV subtests, and these effects were exacerbated among children with higher levels of Proteobacteria (Appendix A). No species (Appendix A) or functional pathway (Appendix A) modified the exposure–outcome relationship. Adjusting for maternal IQ did not meaningfully change the interpretation of results (Appendix A).

## 4. Discussion

In this preliminary study of the potential role of the microbiome as a modifier of the associations between prenatal exposure to non-prescription consumer products and neurodevelopment, we found suggestive evidence that certain childhood gut bacteria may modify the association between caffeine/acetaminophen and cognitive development. Additionally, prenatal acetaminophen exposure was associated with alterations in the gut microbiome in childhood. These findings, if confirmed in larger and more diverse cohorts, may explain differences in the prior literature and suggest therapeutic interventions in cases where exposure is unavoidable.

Prenatal acetaminophen showed a trend of association with lower microbial diversity in childhood, accompanied by decreased relative abundance of Firmicutes and increased relative abundance of Actinobacteria. Higher taxonomic diversity is generally thought to indicate a healthier system in adults, as it contains greater functional diversity to complement the host’s response to exposures [46]. Infants have low taxonomic diversity that increases with the introduction of solid foods [47]. Similarly, vaginally delivered infants have relatively low levels of Firmicutes and high levels of Actinobacteria compared to later samples [26]. It is possible that prenatal exposure to acetaminophen delays the maturation of the gut microbiome to a more adult-like composition. This hypothesis should be explored with more regular exposure assessment and microbiome sampling. In addition, the reason that no species level associations were detected could be due to the relatively small sample size. Future investigations with a larger sample size could help elucidate differences at the species level. Prenatal acetaminophen was associated with an increased relative abundance of genes in the pathway of succinate fermentation to butanoate. Metatranscriptomic or metabolomic analyses could inform whether there are functional differences as a result of differences in the relative abundance of this pathway and whether there are downstream health consequences.

In contrast, we did not observe any associations between prenatal caffeine exposure and gut microbial features. While no epidemiologic studies of changes in the gut microbiome resulting from caffeine exposure have been reported, rats who consumed caffeinated coffee with their chow experienced an increase in Enterobacteriaceae and *Clostridium leptum* [27]. These results may be due to other chemical compounds found in coffee (e.g., antioxidants), but another study found that rats treated with high, repeated doses of caffeine (not reflective of the fetal experience) had increased *Lactobacillus* compared to untreated rats [28]. It is possible that our study was underpowered to detect these changes or that caffeine exposure in the naïve gut is not relevant to later microbiome composition. Although we did not observe differences in species or phylum relative abundance related to caffeine exposure, several gene pathways had increased relative abundance with increasing exposure. For example, concurrent exposure was associated with an increased relative abundance of methionine synthesis genes. This may be in response to increased methyl group availability, as caffeine metabolism requires demethylation. In vitro models may inform the underlying mechanism of this and other associations.

The associations of both acetaminophen and caffeine with cognitive domains were modified by the relative abundance of Proteobacteria in the childhood gut. Proteobacteria includes many notorious pathogens, including *Escherichia, Shigella*, and *Salmonella*, all in the Gammaproteobacteria. An elevated relative abundance of the phylum is associated with worse cognitive performance or impairment in clinical [48,49], population-based [50,51], and animal studies [52], in addition to being considered a marker of gut dysbiosis [53]. Our findings suggest that in addition to the direct negative relationship between Proteobacteria relative abundance and cognitive scores observed in prior studies, it may modify the associations of caffeine and acetaminophen with cognition in an adverse manner. In the case of acetaminophen, Proteobacteria weakened the beneficial impact of acetaminophen on short-term memory, general knowledge, and our measure of overall cognition. In contrast, in the case of caffeine, Proteobacteria exacerbated the adverse impact of caffeine on memory and cognition. This may suggest that Proteobacteria acts through similar pathways (e.g., immune response or oxidative stress) as caffeine and acetaminophen in their associations with cognition [53,54,55].

There is substantial interest in the microbiome as a mediator of health effects. We did not uncover bacterial features at 6–7 years that indicate they may mediate the caffeine/acetaminophen-cognition association (i.e., no feature associated with an exposure that is also associated with cognition in our cohort). However, our study is underpowered to detect small effects and does not discount the possibility that the microbiome mediates these or other associations between chemicals and health outcomes. Importantly, our findings highlight the need to consider whether features of the microbiome, especially Proteobacteria, can alter host response to exogenous exposures, including pharmaceuticals. Because Proteobacteria are overrepresented in the microbiomes of patients with inflammatory bowel diseases [56] and cognitive decline [50,51], compared to healthy controls, this could be particularly important when designing precision medications for these outcomes.

The associations of caffeine and acetaminophen with the microbiome may differ when considering prenatal and mid-childhood exposure due to windows of microbial susceptibility to these exposures or due to differences in the exposure matrix. Fecal concentrations of exposure represent the environment of the gut bacteria more directly than other biomarkers, thus are an intriguing option for microbiome studies. However, meconium begins to accumulate in the second trimester and captures cumulative exposure over the last six months of pregnancy [57], while stool accumulates over a much shorter period (~24 h), capturing more acute exposures [58]. This is captured by the differences in caffeine concentrations between meconium and stool in our study. In addition to differences arising from acute vs. chronic exposures, meconium captures exposure in the naïve gut, which may have a larger impact on long-term microbiome composition by altering the suitability of the gut environment for early colonizers of the intestine. Future animal studies with repeated exposure or epidemiological studies with more frequent exposure assessment could inform windows of susceptibility of the pediatric gut microbiome.

Our study should be considered within the context of its limitations. Due to the small sample size, we could adjust for a limited number of confounders. We prioritized confounders of the association between exposure and the microbiome and selected those that likely have the strongest confounding effects. Ideally, we would control for factors that confound the association between caffeine exposure and neurodevelopment and factors that confound the association between the microbiome and neurodevelopment. Our exposure assessment assay only quantified concentrations of the parent compounds (acetaminophen/caffeine) and not their metabolites. Both caffeine and acetaminophen are known to cross the placenta, whereas the metabolites are more rapidly excreted, and the fetus has limited capacity to metabolize either compound [23,59,60]. Therefore, we captured the majority of fetal exposure from the second trimester (when meconium begins accumulating) to birth. An assessment of the early-life microbiome in relation to prenatal acetaminophen/caffeine exposure may reveal transient disturbances in the microbiome that relate to neurodevelopment during a sensitive window. Our cross-sectional exposure data reduce the possibility that our findings result from correlations between prenatal and childhood exposures. There is potential for reverse causation due to our fecal samples and cognitive measures being collected at the same age.

This novel study also has many strengths. This is one of the first studies to examine the effects of chemicals on the developing microbiome and the first to examine prenatal pharmaceutical exposure. As such, it is particularly interesting and innovative that we measured exposure in the meconium, a matrix that represents the naïve gut environment that bacteria colonize in early life. Further, we used a quantitative measure of exposure to acetaminophen and caffeine rather than relying on maternal reports of intake during pregnancy. Self-report may be under-reported due to the perception of caffeine consumption during pregnancy as a risk behavior or subject to recall bias in cases where the exposure data are collected retrospectively. The longitudinal design and temporality of associations remove the possibility of reverse causation, a common concern of microbiome epidemiology studies. Our use of metagenomic sequencing to profile the childhood microbiome allowed us to examine many aspects of the microbiome, including functional potential, which may result from exposure or modify the exposure–outcome relationship.

## 5. Conclusions

In conclusion, we identified an association between prenatal exposure to acetaminophen and gut microbiome composition in mid-childhood. Larger studies are needed to confirm these results and examine the mechanisms by which in utero exposures affect long-term gut microbiome development. Specifically, studies should collect perinatal maternal and infant microbiome samples to determine whether exposures are associated with the maternal taxa, which are then passed to the infant or if exposures act on the microbiome by affecting what taxa are able to colonize the naïve gut. In addition, we found evidence that certain bacteria may modify the relationships between exogenous exposures and health outcomes, a possible explanation for discrepancies in prior studies on these exposures. The potential for the microbiome to confer resilience or susceptibility to toxic exposures should be explored more fully, including varying exposures, windows of exposure, and outcomes. Overall, this study positions the microbiome as a critical, modifiable feature for future environmental health studies.

## Figures and Tables

**Figure 1 ijerph-19-09357-f001:**
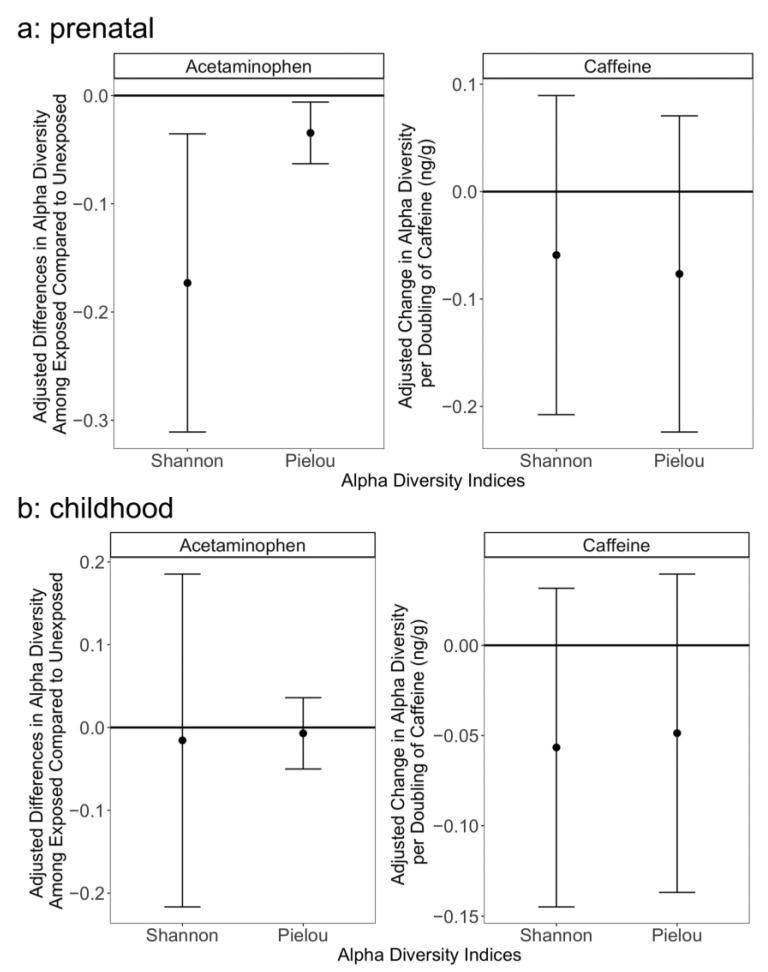
Association between acetaminophen/caffeine concentrations in (**a**) meconium (n = 49) and (**b**) 6–7-year-old stool (n = 85) and alpha diversity. Diversity was measured with Shannon Diversity Index and Pielou Evenness Index. Whiskers represent 95% confidence intervals. Models are adjusted for whether the child was ever breastfed, sex, mode of birth, and family income.

**Figure 2 ijerph-19-09357-f002:**
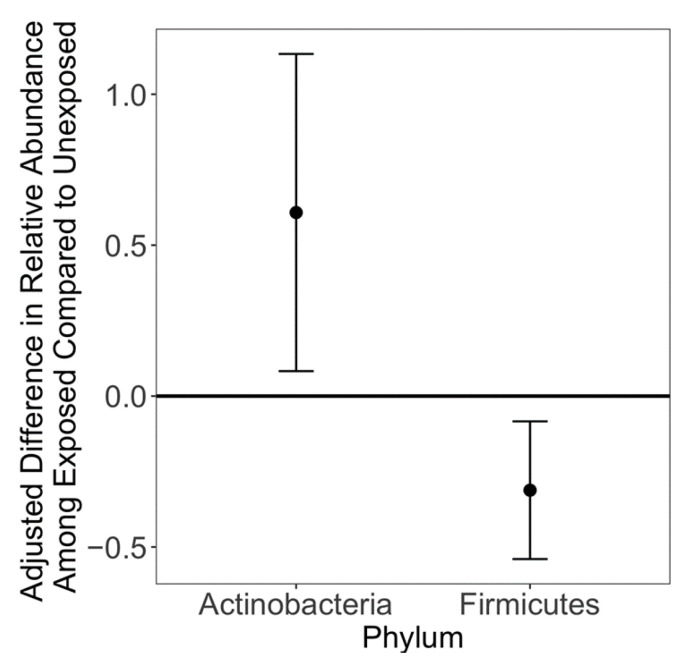
Association between prenatal exposure to acetaminophen and phyla (n = 49). Whiskers represent 95% confidence intervals. Models adjust for whether the child was ever breastfed, sex, mode of birth, and family income.

**Figure 3 ijerph-19-09357-f003:**
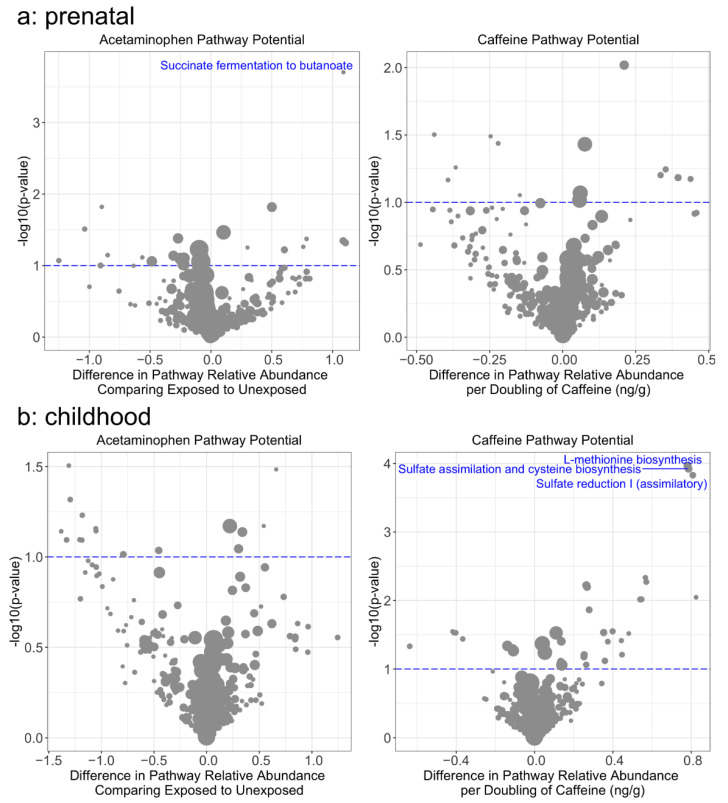
Association between (**a**) prenatal (n = 49) and (**b**) childhood exposure (n = 85) to caffeine or acetaminophen and relative abundance of functional pathways. Effect estimates for caffeine are expressed per doubling of exposure. Effect estimates for acetaminophen are comparing exposed to unexposed. Models are adjusted for whether the child was ever breastfed, sex, mode of birth, and socioeconomic status. Point size is proportional to average relative abundance of the represented pathway. Pathways with an FDR *q*-value < 0.1 are labeled. Full names: L-methionine biosynthesis = Superpathway of L-methionine biosynthesis (by sulfhydrylation); Sulfate assimilation and cysteine biosynthesis = Superpathway of Sulfate assimilation and cysteine biosynthesis.

**Figure 4 ijerph-19-09357-f004:**
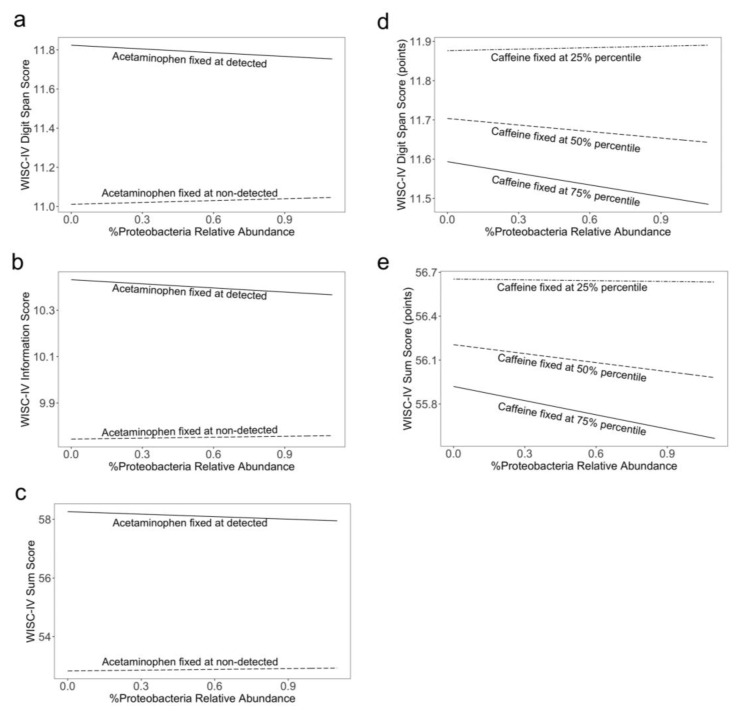
Interaction between meconium concentrations of caffeine and acetaminophen with Proteobacteria (n = 49). Models are adjusted for whether the child was ever breastfed, sex, mode of birth, and socioeconomic status. In (**a**–**c**), solid lines show the association when acetaminophen is detected, and dashed lines show the association when acetaminophen is not detected. In (**d**–**e**), solid lines show the association when caffeine is at the 75th percentile, dashed lines show when caffeine is at the 50th percentile, and dash-dot lines show when caffeine is at the 25th percentile. Subfigures (**a**,**d**) show the association between Proteobacteria and WISC-IV Digit Span scores; subfigure (**b**) shows the association between Proteobacteria and WISC-IV Information scores; subfigures (**c**,**e**) show the association between Proteobacteria and WISC-IV summary scores.

**Table 1 ijerph-19-09357-t001:** Characteristics of the GESTE meconium analytic population (N = 49), cross-sectional analytic population (n = 85), population with meconium (N = 197), and overall population (N = 365) at 6–7-year-old follow-up [expressed as mean ± SD or n (%), unless specified otherwise].

	Full Cohort (n = 365)	Population with Meconium (n = 197)	Meconium Analytic Population (n = 49)	Cross-Sectional Analytic Population (n = 85)
Family Characteristics				
Maternal Age at Recruitment (years)	29.1 ± 4.44	29.4 ± 4.45	29.1 ± 3.85	28.9 ± 4.09
Maternal Pre-pregnancy Body Mass Index (kg/m^2^)				
Available	25.3 ± 5.95	25.4 ± 5.85	25.5 ± 6.30	25.2 ± 6.01
Missing	68 (18.6)	13 (6.6)	12 (24.5)	16 (18.8)
Family Income (CAD ^a^)				
Available	70,800 ± 41,200	71,600 ± 47,300	69,600 ± 34,600	93,900 ± 47,100 ^b^
Missing	20 (5.5)	14 (7.1)	4 (8.2)	6 (7.1) ^b^
Parity				
Nulliparous	205 (56.2)	113 (57.4)	26 (53.1)	46 (54.1)
Parous	158 (43.3)	83 (42.1)	23 (46.9)	39 (45.9)
Missing	2 (0.5)	1 (0.5)	0 (0)	0 (0)
Birth Characteristics				
Gestational Age (weeks)	39.1 ± 1.43	39.1 ± 1.44	39.4 ± 1.09	39.4 ± 1.18
Birth Mode				
Vaginal	299 (81.9)	162 (82.2)	40 (81.6)	68 (80)
Caesarean Section	66 (18.1)	35 (17.8)	9 (18.4)	17 (20)
Child Sex				
Male	199 (54.5)	107 (54.3)	25 (51.0)	44 (51.8)
Female	166 (45.5)	90 (45.7)	24 (49.0)	41 (48.2)
Child Birthweight (g)	3400 ± 486	3370 ± 484	3440 ± 419	3460 ± 445
Breast Feeding Status				
Ever breastfed	284 (77.8)	159 (80.7)	43 (87.8)	68 (80.0)
Never breastfed	65 (17.8)	30 (15.2)	4 (8.2)	14 (16.5)
Missing	16 (4.4)	8 (4.1)	2 (4.1)	3 (3.5)
Neurological Outcomes				
WISC-IV ^c^: Block Design	9.54 ± 2.92	9.43 ± 2.88	10.4 ± 2.99	10.3 ± 2.98
WISC-IV: Coding	10.5 ± 2.94	10.5 ± 2.92	10.8 ± 2.56	11.1 ± 2.67
WISC-IV: Digit Span	9.40 ± 2.58	9.39 ± 2.61	10.0 ± 2.26	9.81 ± 2.18
WISC-IV: Information	9.55 ± 2.29	9.37 ± 2.28	9.45 ± 2.17	9.81 ± 2.23
WISC-IV: Vocabulary	10.4 ± 2.70	10.1 ± 2.76	10.3 ± 3.29	10.6 ± 2.97
WISC-IV Summary Score	49.4 ± 8.36	48.9 ± 8.17	51.0 ± 7.70	51.7 ± 7.86
QTAC ^d^	61.0 ± 8.91	61.6 ± 8.32	62.2 ± 8.20	62.1 ± 8.11
Maternal Intelligence Quotient				
>95 percentile			30 (61.2)	
≤95 percentile			19 (38.8)	
Exposure of Interest				
Caffeine (median [25th percentile, 75th percentile]; ng/g meconium)	-	399 [2.82, 5170]	390 [15.3, 3110]	24.7 [0.184, 231] ^e^
Acetaminophen				
Detected	-	100 (50.8)	20 (40.8)	7 (8.2) ^b^
Not Detected	-	97 (49.2)	29 (59.2)	78 (91.8) ^b^

^a^ Canadian dollars. ^b^ Family income at year 6–7 follow-up. ^c^ Wechsler Intelligence Scale for Children, 4th edition. ^d^ Questionnaire sur le Trouble de L’Acquisition de la Coordination. ^e^ Exposure concentrations in year 6–7 children stool.

## Data Availability

Epidemiologic data are not available due to their sensitive and potentially identifiable nature. Requests to work with the GESTation and Environment cohort should be addressed to Larissa Takser.

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
