# Peer review of "In Utero Exposure to Caffeine and Acetaminophen, the Gut Microbiome, and Neurodevelopmental Outcomes: A Prospective Birth Cohort Study"

_ijerph, 2022, doi:10.3390/ijerph19159357_

Round 1

Reviewer 1 Report

Overall, this study adds new information about the association between prenatal acetaminophen and caffeine exposure and the gut microbiome in childhood, an understudied time point in microbiome literature. This study was appropriately conducted and the manuscript is well-written, however it could be improved with the following changes.

Main text:

I am having trouble understanding the context of this cohort. Where is the cohort based? What’s the range of income, or maternal education? It’s difficult to understand the breadth of SES and education level as a baseline for understanding neuro outcomes and access to prenatal care (re: caffeine and acetaminophen recommendations) in this cohort. While the cohort has been described elsewhere, it would be beneficial to add these key pieces of context to this manuscript.

On line 262-264 you say “Our findings support a direct negative relationship between Proteobacteria relative abundance and cognitive scores,” but I don’t see that analysis anywhere. Only the links between exposures and microbiome, exposures and cognitive scores, and effect modification by Proteobacteria.

While I agree that the longitudinal design removes possibility of reverse causality in your primarily analysis, your microbiome and neuro outcomes were measured at the same time, thus there is potential for reverse causality in your modification effect estimates. For instance, kids with higher cognitive performance (associated with caffeine and acetaminophen levels) may eat differently than kids with lower cognitive performance, which may alter the composition of their microbiome and their relative abundance of Proteobacteria.

There is little if any mention in the discussion of the metabolic pathways you found to be associated with the exposures. Adding some discussion about those findings would help readers better understand them.

Tables and Figures:

Some of the acronyms in the tables have not been defined.

Add timepoint labels to figures 1, 3, and 4 so readers can figure out which samples you are referring to on first glance.

Labeled pathways on figure 3 could be improved. Standardize the label size and/or put the label next to the corresponding dot on the plot.

Figure 4 is blurry, maybe an artifact of the review process, but also fonts are too small to read easily.

Because you have different sample sizes at different timepoints, it would be helpful to include the analytical sample size in each figure legends.

Reviewer 2 Report

The title of the manuscript is somewhat misleading.  I was hoping to see more discussion of neurodevelopmental outcomes.  That exposure to acetaminophen and caffeine are supported, the association with neurodevelopmental outcomes is a bit vague.
